# Study on the Synthesis of Mn_3_O_4_ Nanooctahedrons and Their Performance for Lithium Ion Batteries

**DOI:** 10.3390/nano10020367

**Published:** 2020-02-20

**Authors:** Yueyue Kong, Ranran Jiao, Suyuan Zeng, Chuansheng Cui, Haibo Li, Shuling Xu, Lei Wang

**Affiliations:** Department of Chemistry, Liaocheng University, Liaocheng, Shandong 252059, China; 15953766815@163.com (Y.K.); JRR1175@163.com (R.J.); cuichuansheng@lcu.edu.cn (C.C.); lihaibo@lcu.edu.cn (H.L.); xushuling@lcu.edu.cn (S.X.)

**Keywords:** Mn_3_O_4_ nano-octahedrons, hydrothermal reduction method, structure characterization, electrochemical property, lithium-ion batteries

## Abstract

Among the transition metal oxides, the Mn_3_O_4_ nanostructure possesses high theoretical specific capacity and lower operating voltage. However, the low electrical conductivity of Mn_3_O_4_ decreases its specific capacity and restricts its application in the energy conversion and energy storage. In this work, well-shaped, octahedron-like Mn_3_O_4_ nanocrystals were prepared by one-step hydrothermal reduction method. Field emission scanning electron microscope, energy dispersive spectrometer, X-ray diffractometer, X-ray photoelectron spectrometer, high resolution transmission electron microscopy, and Fourier transformation infrared spectrometer were applied to characterize the morphology, the structure, and the composition of formed product. The growth mechanism of Mn_3_O_4_ nano-octahedron was studied. Cyclic voltammograms, galvanostatic charge–discharge, electrochemical impedance spectroscopy, and rate performance were used to study the electrochemical properties of obtained samples. The experimental results indicate that the component of initial reactants can influence the morphology and composition of the formed manganese oxide. At the current density of 1.0 A g^−1^, the discharge specific capacity of as-prepared Mn_3_O_4_ nano-octahedrons maintains at about 450 mAh g^−1^ after 300 cycles. This work proves that the formed Mn_3_O_4_ nano-octahedrons possess an excellent reversibility and display promising electrochemical properties for the preparation of lithium-ion batteries.

## 1. Introduction

In the past few years, lithium-ion batteries (LIBs) have increasingly held attention owing to their great potential in powering and in hybrid electric vehicles [1]. Thanks to their flexible structure and excellent chemical-physical properties, manganese-based materials (such as Mn_3_O_4_, MnO_2_, and MnOOH) have been applied in the areas of energy storage and catalysis and attracted more interest [2,3]. Compared with other transition metal oxides, Mn_3_O_4_ nanostructure has higher theoretical specific capacity and lower operating voltage [4]; it has been considered as one of the promising electrode materials for the preparation of LIBs [5,6]. However, the low electrical conductivity and large volume change during charge/discharge processes decrease its specific capacity, render rapid capacity loss, and then restrict its applications in the energy conversion and energy storage [5,7]. 

Now, Mn_3_O_4_ materials with micro/nanosizes have been synthesized by a variety of methods, such as the hydrothermal method [8,9], solvothermal method [10], precipitation method [11], chemical decomposition [12], sputtering [13], electrodeposition [14], thermal decomposition [15], and so on. Mn_3_O_4_ nanoparticles with different morphologies have been successfully prepared. Generally, Mn_3_O_4_ materials were synthesized using commercial materials such as manganese sulphate [16], nitrate [17], acetate [18,19,20], and KMnO_4_ [21,22,23,24]. Han et al. produced Mn_3_O_4_ nano-particles using solvent evaporation technique from manganese acetyl acetonate [25]. Yue and Gu et al. synthesized hollow nano-spheres and hierarchical microspheres of MnO_2_ materials using KMnO_4_ and MnSO_4_ via the hydrothermal method, and to obtain hollow nanospheres and hierarchical micro-spheres Mn_3_O_4_ [26,27]. Raj et al synthesized Mn_3_O_4_ by the reduction of KMnO_4_ solution with hydrazine hydrate [21]. Thus, substantial efforts for the preparation of nanosized Mn_3_O_4_ have been performed to improve the electrochemical performance [3,28]. The LIBs prepared by Mn_3_O_4_ tetragonal bipyramids display high initial discharge capacity (1141.1 mAh g^−1^), excellent reversible specific capacity (822.3 mAh g^−1^), good rate performance, and higher coulombic efficiency [8]. Although much progress has been achieved, an effective, low-cost, and safe method to prepare Mn_3_O_4_ nano-octahedrons with uniform size to improve their electrochemical properties is highly desirable.

In this study, octahedron-like Mn_3_O_4_ nanostructures were prepared by the one-step hydrothermal reduction method. The nature of the formed manganese oxides can be effectively controlled by the initial reactant composition. A higher volume ratio of N-dimethylformamide (DMF) to KMnO_4_ in the initial reaction solution contributes to the formation of pure Mn_3_O_4_ nano-octahedrons and the improvement of the capacity stability of the formed product.

## 2. Materials and Methods

### 2.1. Chemicals

Potassium permanganate (KMnO_4_), N, N-dimethylformamide (DMF), anhydrous ethanol, acetylene black, polyvinylidene fluoride, N-methyl-pyrrolidone, metallic lithium, ethylene carbonate, and dimethyl carbonate were of analytical grade and purchased from Sinopharm Chemical Regent Co., Ltd. (Shanghai, China) The water used in the experiments was prepared from a Milli-Q water purification system (≥18 MΩ).

### 2.2. Synthesis of Mn_3_O_4_ Nano-Particles

In this work, Mn_3_O_4_ nano-particles could be prepared by a KMnO_4_ hydrothermal reduction process. Firstly, 20 mL DMF was added into 20 mL 0.02 mol L^−1^ KMnO_4_ aqueous solution under stirring, and the formed mixture was transformed into a Teflon-sealed autoclave. Then, the reaction system was heated to 150 °C and held for 9 h. Finally, the precipitates were collected by centrifugal separation, washed thoroughly with water and ethanol, and dried for testing. The obtained particles are Mn_3_O_4_ nano-octahedrons.

As 10 mL DMF was mixed with 30 mL 0.02 mol L^−1^ KMnO_4_, and used as initial reactants for the above hydrothermal process, the obtained product contained MnOOH nano-columns and Mn_3_O_4_ nano-octahedrons and was named MnOOH/Mn_3_O_4_.

### 2.3. Material Characterization

The composition and phase of the obtained product were detected by an Ultima IV multipurpose X-ray diffraction system (XRD, Rigaku Corporation, Tokyo, Japan) using Cu Kα radiation (λ = 0.15406 nm). The morphology of formed sample was determined using a Sirion 200 field emission scanning electron microscope (FE-SEM, FEI Company, Eindhoven, Netherlands), while the chemical composition was characterized by an INCA energy dispersive spectrometer (EDS, Oxford Instruments, Oxon, England, UK). The lattice parameters and lattice images of as-prepared samples were tested by a Talos F200X field-emission high resolution transmission electron microscopy (FE-HRTEM, Thermo Fisher Scientific, Waltham, MA, USA). The surface constituents and element valence state of formed products were detected with an ESCLAB MKII X-ray photoelectron spectrometer (XPS, VG Scientific, Waltham, MA, USA). The binding energy of the obtained XPS spectra was calibrated by the C (1s) peak at 284.6 eV. The infrared absorption spectra were measured on a 6700 Fourier transformation infrared (FT-IR) spectrometer (Thermo Fisher Scientific, Waltham, MA, USA).

### 2.4. Electrochemical Measurements

In order to study the discharge/charge performances of Mn_3_O_4_ nano-octahedrons and MnOOH/Mn_3_O_4_, working electrodes were fabricated. The as-prepared manganese oxide, carbon black, and poly(vinylidene fluoride) were firstly blended in a weight ratio of 70:20:10. Then, N-methyl-pyrrolidone was added to prepare a paste. The formed slurry was homogeneously coated onto a copper foil, dried, and then used as anode. The assembly process of coin cells was conducted in a Mikrouna glove box. Metallic lithium and a polypropylene membrane with Celgard 2400 micropores were used as separator film and cathode, and 1 mol L^−1^ LiPF_6_ was selected as the electrolyte.

Galvanostatic charge–discharge properties and rate capabilities of the formed LIBs were conducted on a CT2001A LAND battery test system. Cyclic voltammetry (CV) and electrochemical impedance spectroscopy (EIS) were determined by a Zennium-pro electrochemical workstation.

## 3. Results

### 3.1. Microstructure and Morphology 

The microstructures of the obtained products were characterized by XRD, as shown in Figure 1. For the product prepared from 30 mL 0.02 mol L^−1^ KMnO_4_ and 10 mL DMF, two phases that corresponded to MnOOH (JCPDS 74-1842) and Mn_3_O_4_ (JCPDS 24-0734) could be ascertained (Figure 1a). The diffraction peaks that correspond to the crystal planes of hausmannite Mn_3_O_4_ were labelled. The second phase represents an orthorhombic MnOOH, and the main diffraction peak at 26.30 degree corresponds to the crystal plane (110) of orthorhombic MnOOH. Other diffraction peaks at 34.15, 37.20, 39.80, 40.60, 55.10, and 71.70 degrees could be attributed to the (020), (101), (120), (200), (211), and (301) crystal planes of orthorhombic MnOOH, respectively.

For the product prepared from 20 mL 0.02 mol L^−1^ KMnO_4_ and 20 mL DMF, all the diffraction peaks can be indexed to the tetragonal hausmannite Mn_3_O_4_ (JCPDS 24-0734) (Figure 1b). The sharp peaks imply that the product possesses high crystalline quality. The results indicate that an increased volume ratio of DMF in the initial reactant solution contributes to the formation of pure Mn_3_O_4_ nanooctahedrons.

The morphology of the obtained sample was investigated by typical FE-SEM, as shown in Figure 2. For the mixture of MnOOH and Mn_3_O_4_, it contains two different nanostructures (Figure 2a–c). The long thin nano-columns are about 4.0 μm in length and 200 nm in diameter. Their aspect ratios are about 20. The nano-octahedrons are about 400 nm in an edge length. For the pure Mn_3_O_4_, nano octahedrons with uniform size, clear boundary, and point angle can be seen, and the mean edge length of nano-octahedron is about 400 nm (Figure 2d–f).

On the basis of the XRD patterns, we can assume that the formed nano-octahedrons correspond to Mn_3_O_4_, while the nano-columns can be attributed to the MOOH structure. FE-SEM images also demonstrate that the increased volume ratio of DMF in the initial reactant solution can contribute to the formation of pure Mn_3_O_4_ nano-octahedrons.

EDS was applied to determine the chemical composition of obtained samples, it confirms the existence of Mn and O. For the MnOOH/Mn_3_O_4_, the atomic ratio of Mn to O is 35.45:64.55 (Figure 3a), and the atomic ratio of Mn/O is less than 3:4, but larger than 1:2, which also suggests that the obtained product is the mixture of MnOOH and Mn_3_O_4_. For the Mn_3_O_4_ nano-octahedrons, the atomic ratio of Mn to O is 41.42:58.58, which is close to 3:4 and certifies the formation of Mn_3_O_4_.

In the hydrothermal process, DMF plays an important role in chemical reaction and in oriented self-assembly. Firstly, DMF is one kind of Lewis base; it is easy to combine with H^+^ in aqueous solution. So the initial reaction solution that contains DMF and KMnO_4_ is alkaline. At the initial stage of the reaction, Mn^3+^ or MnOOH nuclei are formed as a sequence of the reduction of KMnO_4_. Plenty of −OH groups in the water molecules would contribute to the formation of MnOOH and restrict the phase transformation of MnOOH to Mn_2_O_3_. At the same time, DMF was oxidized to *N*,*N*-dimethyl carbamic acid. The formation of N,N-dimethyl carbamic acid will change the pH value of the reaction solution, and act as a binding agent to control the nucleation and growth of crystals [29,30]. The formed nano-crystallites will accumulate and grow into lamellar MnOOH nano-structures. However, the formed lamellar nano-structures are just intermediates. As the reaction time is prolonged, lamellar MnOOH nano-structures will be re-dissolved and then grow into small nanowires as a result of the anisotropic growth [31,32]. The small nanowires prefer to combine with each other and grow into larger and longer nano-columns by the oriented attachment along their side surfaces [33]. As Mn^2+^ partially formed by further reduction of MnOOH, some octahedron-like Mn_3_O_4_ nanostructures began to occur. So the product obtained from the high volume ratio (3:1) of 0.02 mol L^−1^ KMnO_4_ to DMF is the mixture of MnOOH nano-columns and Mn_3_O_4_ nano-octahedrons.

As the volume ratio of 0.02 mol L^−1^ KMnO_4_ to DMF decreases to 1:1, the MnOOH nano-column will be reduced into Mn^2+^ by DMF. MnOOH nano-columns dissolve and disappear with the formation of octahedron-like nanostructures. It should be noted that DMF can coordinate with MnO^-^ and regulate the growth rate of Mn_3_O_4_ crystal by binding crystal facets [34,35]. The formed complex provides a suitable micro-environment for the formation of Mn_3_O_4_ nano-octahedrons [36,37]. At the same time, the pH value of the reaction system is important to the formation of uniform Mn_3_O_4_ nano-octahedrons. The increased volume ratio of DMF in the initial reactant solution means a higher pH value, which facilitates the growth along the <100> direction of Mn_3_O_4._ Furthermore, the ratio of crystal growth rate in the <100> direction to that in the <111> direction dominates the lattice structure and morphology of formed Mn_3_O_4_ [38]. Because the <111> plane of Mn_3_O_4_ has the lowest energy, eight stable crystal planes arrange on the surface of nano-octahedron. Finally, all of the products transformed into uniform nano-octahedrons with smooth surfaces. Under hydrothermal conditions, the coexistence of Mn^2+^ and Mn^3+^ facilitates the formation of the Mn_3_O_4_ crystal. So, the nature of formed manganese oxides can be effectively controlled by the initial reactant composition. Pure Mn_3_O_4_ nano-octahedrons can be synthesized by the higher volume ratio of DMF to KMnO_4_ (1:1).

### 3.2. HR-TEM and Elemental Mapping

TEM can be used to observe the morphology of obtained manganese compounds. For the MnOOH/Mn_3_O_4_, nano-columns and nano-octahedrons can be identified, which is identical to FE-SEM images. The edge lengths of as-prepared nano-octahedrons are about 400 nm, while the formed nanocolumns are about 100–200 nm in diameter and up to 10 μm in length (Figure 4a,b). The interplanar distances calculated from the HR-TEM images are 0.219 and 0.262 nm (Figure 4c), which correspond to the (111) and (020) planes of monoclinic Mn_3_O_4_, respectively. The elemental mapping images demonstrated that Mn and O components are uniformly distributed all over the Mn_3_O_4_ nano-octahedrons and MnOOH nanocolumns (Figure 4d–f).

For the Mn_3_O_4_ nano-octahedrons, an average rhombic length of 300–500 nm can be found (Figure 4g–h), which is in agreement with the FE-SEM images. The HR-TEM image of an octahedral nanocrystal displays a lattice spacing of 0.236 nm (Figure 4i), and it can be assigned to the (004) plane of hausmannite Mn_3_O_4_ crystal [39]. Elemental dot-mapping images show that Mn and O components are evenly scattered throughout the nano-octahedrons (Figure 4j–l).

### 3.3. XPS Analysis

The surface composition and valence state of obtained products were analyzed by the XPS spectrum, as shown in Figure 5. For the MnOOH/Mn_3_O_4_, the Mn2p spectrum shows that the peaks of Mn2p_3/2_ and Mn2p_1/2_ center at 641.5 and 653.1 eV, respectively (Figure 5a). It can be deconvoluted into two pairs of doublets by fitting with the Gaussian function. The former doublet at 641.3 and 643.35 eV corresponds to Mn^3+^ 2p_3/2_ and Mn^2+^ 2p_3/2_, respectively, while the latter doublet at 652.90 and 654.45 eV could be attributed to Mn^3+^ 2p_1/2_ and Mn^2+^ 2p_1/2_, respectively [40]. Quantitative analysis indicates that the ratio of Mn^2+^/Mn^3+^ is about 1:4. Considering the different valence ratio of the Mn element in the Mn_3_O_4_ and MnOOH, the above calculated value is reasonable. O 1s spectrum can be decomposed into three components (Figure 5b). The first peak at 529.65 eV is the characteristic of lattice oxygen, the second peak at 530.85 corresponds to the O^2−^ ions in oxygen vacancy defect regions, and the third peak at 531.55 eV is related to the adsorbed oxygen or dissociated oxygen [40,41]. The relative percentages of above three O components are 47.3%, 34.5%, and 18.2%.

For Mn_3_O_4_ nano-octahedrons, the Mn 2p XPS spectrum consisted of two spin-orbital lines (Figure 5c). The peaks at 641.50 and 653.20 eV are the characteristic peak of Mn 2p_3/2_ and Mn 2p_1/2_, respectively. A splitting of 11.7 eV occurs between Mn 2p_1/2_ and Mn 2p_3/2_, which is in agreement with the reported value of Mn_3_O_4_ [42,43]. Furthermore, the Mn 2p XPS spectrum could be divided into four components. The peaks at 641.4, 643.4, 652.90, and 654.45 eV can be ascribed to Mn^3+^ 2p_3/2_, Mn^2+^ 2p_3/2_, Mn^3+^ 2p_1/2_, and Mn^2+^ 2p_1/2_, respectively [40]. Quantitative analysis indicates that the ratio of Mn^2+^ to Mn^3+^ is about 0.5, which is consistent with the Mn components with different valence states in Mn_3_O_4_. O 1s spectrum of the formed nano-octahedrons can be resolved into three components with binding energies of 529.75, 530.95, and 532.4 eV (Figure 5d), which correspond to the lattice oxygen of Mn_3_O_4_, the O^2-^ ions of oxygen-vacancy defects within the matrix of Mn_3_O_4_ [41,44], and the adsorbed/dissociated oxygen on the surface of Mn_3_O_4_ [40,41,45]. The relative percentages of above three O components are 55%, 36.7%, and 8.3%. The proportion of lattice oxygen in Mn_3_O_4_ nano-octahedrons is higher than that in the MnOOH/Mn_3_O_4_, while the proportion of adsorbed/dissociated oxygen species is lower. XPS analysis further confirms that the as-prepared nano-octahedrons are pure Mn_3_O_4_.

### 3.4. FT-IR Spectrum Analysis

FT-IR spectra can be used to confirm the formation of Mn_3_O_4_ and MnOOH [40,46]. For the MnOOH/Mn_3_O_4_ (Figure 6a), sharp peaks at 448, 491, and 593 cm^−1^ are the characteristic modes of Mn-O bonds in Mn_3_O_4_ and γ-MnOOH [47]. The peak at 418.6 and 531 cm^−1^ are the characteristic stretching mode and the bending mode of Mn-O bond at the octahedral sites of Mn_3_O_4_, respectively [48]. The peak at 632 cm^−1^ originated from the stretching vibrations of the Mn-O (Mn^3+^) bond at the tetrahedral sites in the Mn_3_O_4_ [10,49]. The peak at 1085 cm^−1^ corresponds to the out-of-plane bending vibration of γ-OH, while the peaks at 1118 and 1151 cm^−1^ arise from the in-plane bending vibrations of δ-2-OH and δ-1-OH, respectively [50]. The above vibration absorption bands are consistent with that of the reported γ-MnOOH and hausmannite Mn_3_O_4_.

For the Mn_3_O_4_ nano-octahedrons (Figure 6b), the characteristic peak at 412 cm^−1^ is associated with the stretching mode of Mn-O-Mn bonds [51], the peak at 446 cm^−1^ comes from the band stretching modes of the octahedral sites [52]. The peak at 594 cm^−1^ is the characteristic band of Mn_3_O_4_, which corresponds to the stretching mode of the Mn-O bond at the tetrahedral site and the stretching mode of the Mn-O bond in an octahedral environment [53]. The peaks at 632 and 530 cm^−1^ originate from the coupling of Mn-O stretching vibrations at the tetrahedral site and at the octahedral site [54]. FT-IR measurement indicated that the obtained products are Mn_3_O_4_ nano-octahedrons.

### 3.5. Electrochemical Property

In this work, the galvanostatic charge–discharge process, CV, EIS, and rate capability were performed to compare the electrochemical properties of as-prepared samples.

During the galvanostatic measurements, the current density was set at 1.0 A g^−1^, while the cutoff voltage was conducted within 0.01–3.0 V. The discharge/charge curves in initial three cycles were presented (Figure 7). For the MnOOH/Mn_3_O_4_ electrode, the discharge voltage drops steeply to 0.34 V. A long discharge voltage plateau at 0.3 V can be found in the first discharge curve, which comes from the reaction between MnOOH/Mn_3_O_4_ and Li [20,55]. In the second discharge process, the main voltage plateau at 0.58 V originates from the reaction between the MnO and Li, the formation of Mn and amorphous Li_2_O [20]. In the third discharge curve, a discharge voltage plateau appears at 0.50 V. For the charging–discharging curves of the Mn_3_O_4_ nano-octahedrons electrode, a short voltage platform at 0.43 V can be found in the first discharging curves, which arises from the reaction occurred between Mn_3_O_4_ and metallic Li. In the second and third cycles, the discharge curves are coincided with each other, and the well-defined discharge voltage platform at 0.45 V originates from the reaction between the MnO and Li. The results indicate that the main discharge platform does not change obviously in the following cycles.

The long-term cyclabilities of as-prepared samples were recorded, as shown in Figure 8. For the MnOOH/Mn_3_O_4_ electrode, the specific capacities in the first discharge and charge processes are 1172.1 and 796.3 mAh g^−1^. The low coulombic efficiency of 67.9% is related to the formation of solid electrolyte interphase (SEI) film [56]. However, the discharge and charge specific capacities decrease to 793.1 and 753.4 mAh g^−1^ in the second cycle. SEI films retard further lithiation of the inner part of Mn specimens and result in decreased discharge/charge capacity [57]. At the same time, the coulombic efficiency increases up to 92.7%. Cyclic performance examination reveals that the charge/discharge specific capacity continuously reduces to 391 mAh g^−1^ in the 73rd cycle. Since then, the discharge/charge specific capacity gradually increases. Finally, the discharge and charge capacity gradually increased to 724.2 and 722.4 mA h g^−1^ in the 300th cycle. Because the shape and size of formed MnOOH nano-columns and Mn_3_O_4_ nano-octahedrons are inhomogeneous, the cracking and fragmentation that originate from the stress during the lithiation reaction will lead to the increased discharge/charge specific capacity. 

For the Mn_3_O_4_ nano-octahedrons electrode, the discharge and charge specific capacities in the first cycle are 971.8 and 592.9 mAh g^−1^, which reveals that the initial discharge capacity of octahedron-like Mn_3_O_4_ nano-structures measured at 1.0 A g^−1^ is higher than the theoretical capacity (937 mAh g^−1^) for the conversion reaction to Mn and Li_2_O [58]. In the second cycle, the discharge and charge specific capacity decreased to 599 and 567.4 mAh g^−1^, while the coulombic efficiency increases up to 94.7%. In the 73rd cycle, the charge/discharge specific capacity reduces to 340 mAh g^−1^. After that, it gradually increases. At the 150th cycle, the reversible specific capacity reaches 443 mAh g^−1^. Since then, the charge/discharge specific capacity basically maintains at 450 mAh g^-1^. Compared with the reported Mn_3_O_4_ nano-rods/wires [59], Mn_3_O_4_ nanotubes [60], Mn_3_O_4_ tetragonal bipyramids [8], Mn_3_O_4_ hierarchical microspheres [27,61], Mn_3_O_4_ hollow microspheres [62], Mn_3_O_4_ nanooctahedra [63], Mn_3_O_4_ nanosheets [64], and spherical-like Mn_3_O_4_-S nanoparticles [65], the formed Mn_3_O_4_ nano-octahedrons present better electrochemical properties.

Because the exposed (011) facets of as-prepared Mn_3_O_4_ nano-octahedrons are highly active, the improved cycling performance could be attributed to their smaller charge transfer resistance. At the same time, the alternating Mn and O atom layers on the exposed (011) facets can speed up the conversion reaction that occurs between Mn_3_O_4_ and Li, facilitate the multi-electron reaction, and ease the formation and decomposition of the amorphous Li_2_O [63].

The details of the lithium insertion–extraction process can be evaluated by the CV method [66,67,68]. In this work, the initial three CV curves were recorded within 0.01–3.0 V at a scan rate of 0.1 mV s^−1^, as shown in Figure 9.

For the MnOOH/Mn_3_O_4_ electrode, two reduction peaks can be found in the first cathodic sweep. The cathodic peaks centered at 0.166 and 0.86 V corresponded to the multistep lithium ion insertion mechanism, which included the reduction of Mn^3+^ to metallic Mn and the decomposition of organic electrolyte. The anodic peak that appeared at 1.21 V originated from the lithium ion extraction and oxidation of Mn to MnO [69]. In the following cycles, the cathodic peaks shifted to 0.30 and 0.41 V, demonstrating the occurrence of irreversible phase transformation after the first cycle. The first reduction peak may be mainly related to the drastic structural modifications that is impelled by lithium [70] and the reduction of MnO to metallic Mn [69], while the second reduction peak corresponds to the irreversible phase transformation during the lithium insertion in the first cycle [20,71,72]. In the initial three anodic sweeps, the oxidation peak is located at 1.2 V. Thus, equations of MnOOH nanocolumns/Mn_3_O_4_ nanooctahedrons electrode used for the rechargeable LIBs can be formulated [73]:(1)$$MnOOH+3Li^{+}+3e^{-}\to Mn+Li_{2}O+LiOH$$
(2)$$Mn_{3}O_{4}+8Li^{+}+8e^{-}\to 3Mn+4Li_{2}O$$
(3)$$Li_{2}O+Mn\to MnO+2Li^{+}+2e^{-}$$

For the Mn_3_O_4_ nano-octahedrons electrode, three reduction peaks at 0.12, 0.88, and 1.06 V can be found in the first cathodic sweep. The intensive peak emerging at 0.12 V may be related to the reduction of Mn_3_O_4_ to MnO [71], the second peak locating at 0.88 V corresponds to the formation of the SEI layer on the electrode surface [69]. The third peak appearing at 1.06 V can be attributed to the reaction between Mn_3_O_4_ and Li during the process of the SEI layer formation [14]. The latter two weak peaks disappeared in the following cycles because they were annihilated in the next cycles [74]. The peak at 1.18 V in the first anodic sweep originated from the oxidation of Mn to MnO and reduction of Li_2_O to Li [75]. In the second cycle, two reduction peaks are located at 0.25 V and 0.39 V, while the oxidation peak is much like that of the first anodic sweep. In the third cycle, the reduction/oxidation peaks are identical to those appeared in the second cycle, which indicate the good reversibility of Mn_3_O_4_ nano-octahedrons electrode [7,70,75]. So the electrochemical reaction that occurs in the first discharge/charge cycle will be summarized as the following equations [14,76,77]:

Initial discharge cycle
(4)$$Mn_{3}O_{4}+Li^{+}+e^{-}\to LiMn_{3}O_{4}\left(1.5-0.5VvsLi^{+}/Li\right)$$
(5)$$LiMn_{3}O_{4}+Li^{+}+e^{-}\to Li_{2}O+3MnO\left(1.5-0.5VvsLi^{+}/Li\right)$$
(6)$$MnO+2Li^{+}+2e^{-}\to Mn+Li_{2}O\left(0.5-0.0VvsLi^{+}/Li\right)$$

Generally,
(7)$$Mn_{3}O_{4}+8Li^{+}+8e^{-}\to 3Mn+4Li_{2}O$$

Initial charge cycle:(8)$$Mn+Li_{2}O\to Mn+2Li^{+}+2e^{-}\left(0.5-3.0V\;vs\;Li^{+}/Li\right)$$

Electrochemical impedance spectroscopy (EIS) was used to investigate the Mn_3_O_4_ electrodes. It can supply related information to compare the electrochemical, electron, and ion transport properties of different electrode materials [78,79,80]. The Nyquist plots as well as the equivalent electrical circuit for the prepared two Mn_3_O_4_ electrodes are shown in Figure 10a. The high frequency intercept on the real axis corresponds to the ohmic resistance (R_s_), which originated from the electrolyte resistance, internal resistance of electrode, and contact resistance [81]. The results indicate that the R_s_ values of the MnOOH/Mn_3_O_4_ and pure Mn_3_O_4_ electrodes are 2.1 and 2.5 Ω, respectively. The semicircle in the high frequency region of Nyquist plot corresponds to charge-transfer resistance (R_ct_). For the Mn_3_O_4_ nano-octahedrons electrode, the smaller diameter suggests that Mn_3_O_4_ nano-octahedrons possess a much lower charge transfer resistance (267.6 Ω) than that of the formed MnOOH/Mn_3_O_4_ (304.2 Ω). In addition, the slope of Nyquist plots in the low-frequency range of the EIS spectra correlates with the diffusion barrier of electrolyte ions. For Mn_3_O_4_ nano-octahedrons electrode, the much larger slope at low frequency intimated the faster electrolyte ion transfer than that in MnOOH/Mn_3_O_4_ electrodes (Figure 10b).

The rate capabilities were examined at the current densities of 0.2, 0.5, 1.0, 2.0, 5.0, and 0.20 A g^−1^. The testing results were applied to compare the electrochemical characteristics of two formed samples, as shown in Figure 11. For the MnOOH /Mn_3_O_4_ electrode, the reversible specific capacities were 627, 516, 439, 368, 303, and 480 mAh g^−1^ (retention: 76.6%; 0.2–0.2 A g^−1^ in the last cycle) (Figure 11a). For the Mn_3_O_4_ nano-octahedrons electrode, the reversible specific capacity gradually decreases with the increase in the testing current density. The specific capacities of 1050, 840, 633, 475, 315, and 958 mAh g^−1^ are obtained. Even at a high current density of 5.0 A g^−1^, Mn_3_O_4_ nano-octahedrons still deliver the specific capacity of 315 mAh g^−1^. After the high current density of 5.0 A g^−1^ was conducted, the charge/discharge specific capacity measured at 0.2 A g^−1^ could be retrieved (retention: 91.2%; 0.2–0.2 A g^−1^ in the last cycle) (Figure 11b), which indicates that Mn_3_O_4_ nano-octahedrons possess an excellent reversibility and display promising electrochemical properties as high capacity LIB electrode materials.

## 4. Conclusions

In this work, KMnO_4_ and DMF were used for the preparation of Mn_3_O_4_ nano-octahedrons by the one-step hydrothemal reduction method. The structure, morphology, and composition of the obtained products were characterized, respectively. Galvanostatic charge–discharge, CV, EIS, and rate performance were used to study the electrochemical properties of obtained samples. The results indicated that the nature of as-prepared manganese oxides could be controlled by tuning the initial reactant composition. As the galvanostatic charge–discharge process was conducted at 1.0 A g^−1^, the initial discharge-charge specific capacities of prepared Mn_3_O_4_ nano-octahedrons are 971.8 and 592.9 mAh g^−1^. After 300 cycles, a discharge/charge specific capacity of 450 mAh g^−1^ can be delivered. The results indicated that the formed Mn_3_O_4_ nano-octahedrons possess outstanding capacity stability and can be used for the preparation of LIBs.

## Figures and Tables

**Figure 1 nanomaterials-10-00367-f001:**
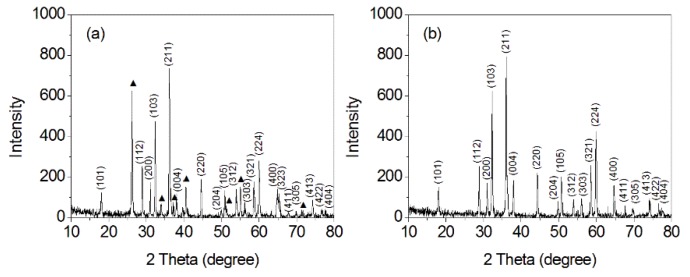
X-ray diffraction (XRD) patterns of obtained products prepared with different volume ratios of 0.02 mol L^−1^ KMnO_4_ to N-dimethylformamide (DMF). (**a**) 3:1, (**b**) 1:1.

**Figure 2 nanomaterials-10-00367-f002:**
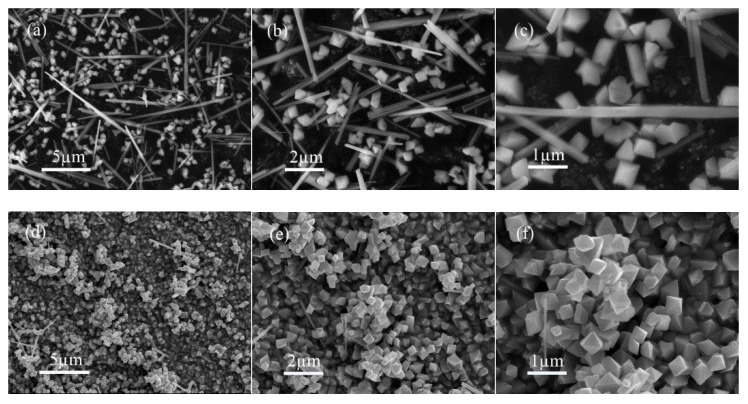
Field emission scanning electron microscope (FE-SEM) images of MnOOH/Mn_3_O_4_ (**a**–**c**) and pure Mn_3_O_4_ (**d**–**f**).

**Figure 3 nanomaterials-10-00367-f003:**
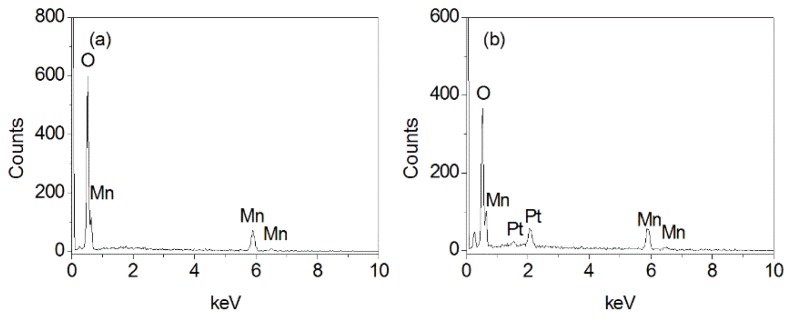
Energy dispersive spectrometer (EDS) of MnOOH/Mn_3_O_4_ (**a**) and Mn_3_O_4_ nanooctahedrons (**b**).

**Figure 4 nanomaterials-10-00367-f004:**
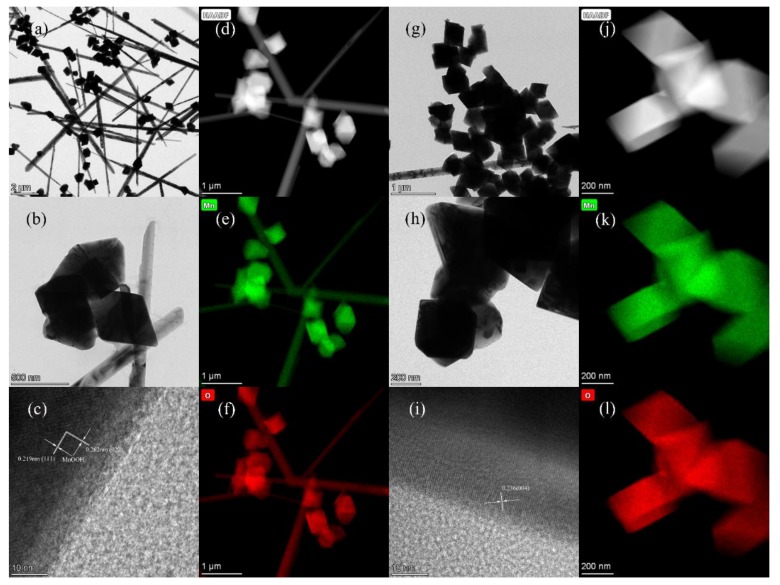
High resolution transmission electron microscopy (HR-TEM) and elemental mapping images of MnOOH/Mn_3_O_4_ (**a**–**f**) and Mn_3_O_4_ nanooctahedrons (**g**–**l**).

**Figure 5 nanomaterials-10-00367-f005:**
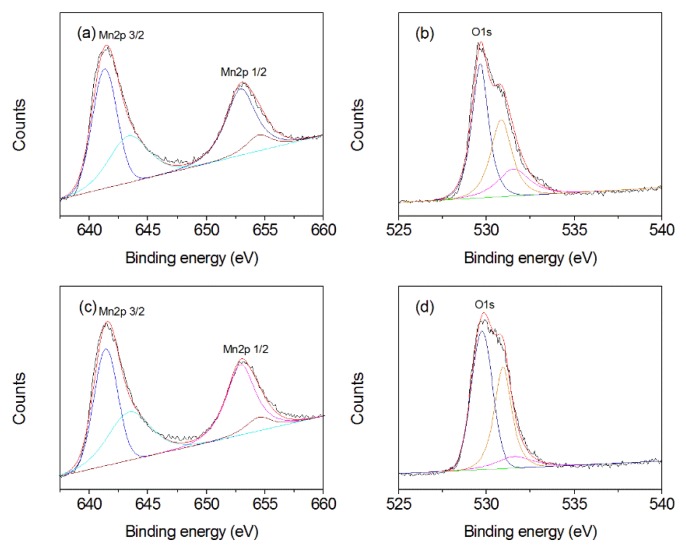
X-ray photoelectron spectrometer (XPS) spectra of MnOOH/Mn_3_O_4_ (**a**,**b**) and Mn_3_O_4_ nanooctahedrons (**c**,**d**).

**Figure 6 nanomaterials-10-00367-f006:**
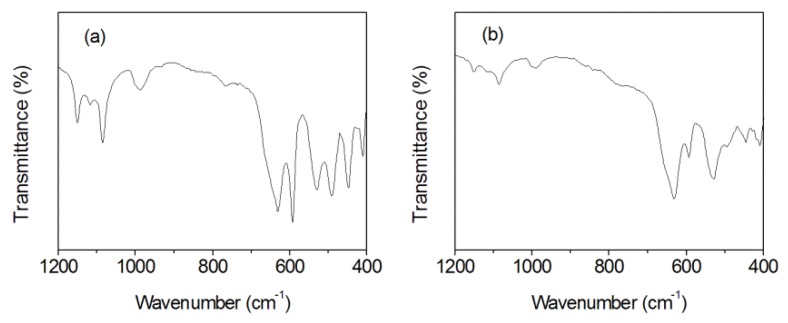
Fourier transformation infrared (FT-IR) of MnOOH /Mn_3_O_4_ (**a**) and Mn_3_O_4_ nanooctahedrons (**b**).

**Figure 7 nanomaterials-10-00367-f007:**
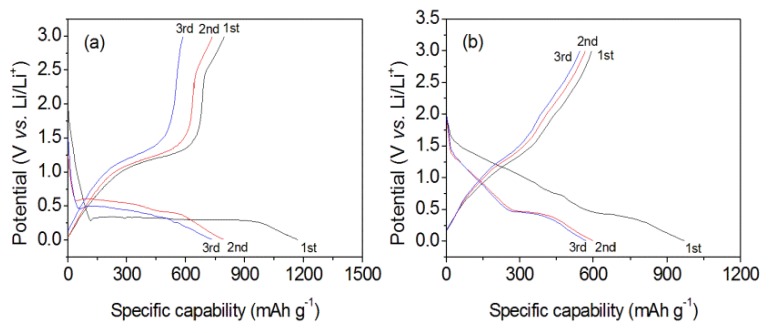
Galvanostatic charge/discharge profiles of MnOOH /Mn_3_O_4_ (**a**) and Mn_3_O_4_ nanooctahedrons (**b**) with a potential window from 0.01 V to 3.0 V at the current density of 1.0 A g^−1^.

**Figure 8 nanomaterials-10-00367-f008:**
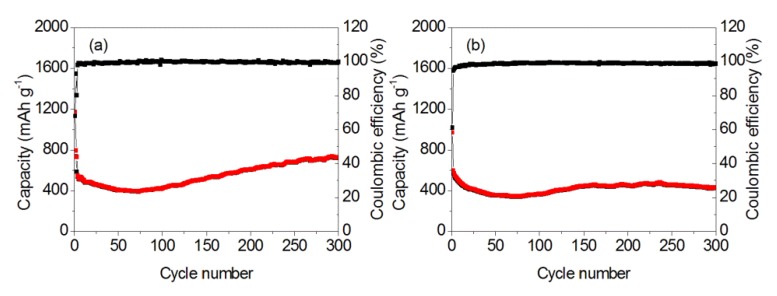
Cycle properties of MnOOH/Mn_3_O_4_ (**a**) and Mn_3_O_4_ nanooctahedrons (**b**) at the current density of 1.0 A g^−1.^

**Figure 9 nanomaterials-10-00367-f009:**
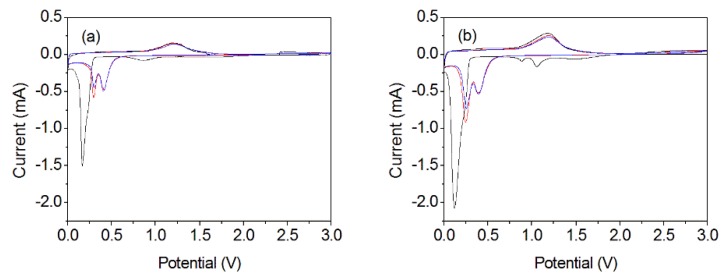
Cyclic voltammetry (CV) curves of MnOOH/Mn_3_O_4_ electrode (**a**) and Mn_3_O_4_ nanooctahedrons electrode (**b**).

**Figure 10 nanomaterials-10-00367-f010:**
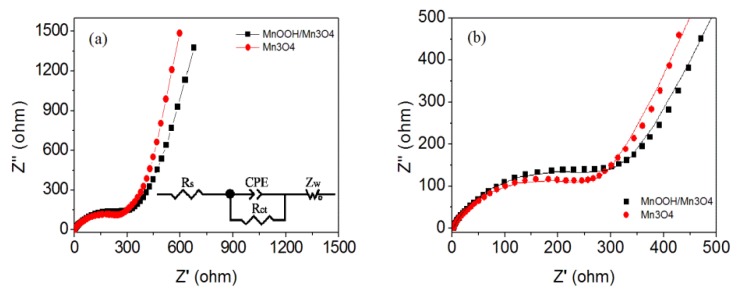
Nyquist plots of MnOOH/Mn_3_O_4_ and Mn_3_O_4_ electrodes (**a**) and the enlarged fitting plots (**b**). The inset shows the corresponding equivalent electrical circuit.

**Figure 11 nanomaterials-10-00367-f011:**
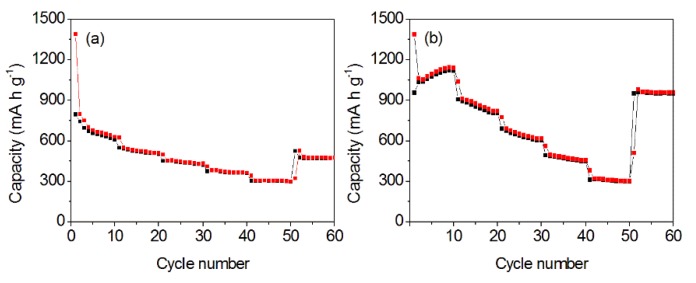
Rate performance of MnOOH/Mn_3_O_4_ electrode (**a**) and Mn_3_O_4_ nanooctahedrons electrode (**b**).

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
