# Peer review of "Study on the Synthesis of Mn3O4 Nanooctahedrons and Their Performance for Lithium Ion Batteries"

_nanomaterials, 2020, doi:10.3390/nano10020367_

Round 1
Reviewer 1 Report
This manuscript deals with the investigation of octahedral-like Mn3O4 crystallites prepared by solvothermal synthesis. The manuscript contains experimental results which could be published to enrich the literature on this issue. Unfortunately, the manuscript is not suitable for the publication. The manuscript lacks of important information, is confusing and some scientific statements of the Authors are questionable.
Corrections/Changes
Improvement of the language. Minor errors in the grammatical English have been found. The errors can be easily corrected.
Improvement of the discussion. Although the Authors have reported references to support their experimental results, this Reviewer has found very few hints of comparison with data previously reported.
Major corrections/changes
Title
The title MUST be changed. Qualitative opinions of the Authors on their own results (“well shaped”, “good performance”) are not scientific at all and give rise to a not good impression on the readers.
Abstract
The Abstract is a very short report of the results obtained during the work. Despite the very remarkable characterization of the samples, this Reviewer has barely found hints of the scientific results obtained by the Authors. The Abstract must be rewritten.
Introduction
Lines 45-50: the Authors must describe the purpose of their work. Any description of the experimental results and/or methods does not belong to the Introduction section but to Conclusions. Please cut out of text.
Experimental Section
Lines 53-60: this part has been poorly described. This section is to be rewritten for the following reasons:
- What is the KMnO4 solution utilized by the Authors? Water, organic one? Was the KMnO4 solution prepared by the Authors or purchased by some company?
- The Authors describe the preparation of the 1:1 sample only but they describe the characterization of the samples 1:1 and 3:1 with no mention in the preparation section;
- This Reviewer has found no mention of the description of the form of the samples utilized for the structural, microstructural, spectroscopic characterization As-prepared? Grounded?
Furthermore, the Authors are strongly recommended to label the prepared samples in order to avoid any repetition of the sentence “For the product prepared from 20 mL 0.02 M KmnO4 and 20 mL DMF” throughout the manuscript and in the Captions for Figures
Results and discussion
This part contains inaccurate statements and conclusions and must be completely rewritten.
The results should be presented in tables (e.g.: IR vibrational modes) in order to improve the quality of the manuscript.
Indeed the reactivity of MnO, Mn2O3, Mn3O4 and MnO2 is associated with the capacity of manganese to form various oxidation states, e.g. redox reaction of Mn2+/Mn3+ or Mn3+/Mn4+, and “oxygen mobility” in the oxide lattice. That also means that the formation of the Mn-oxides is strongly dependent on the pH of the solution.
The Authors prepared their samples starting from Mn(VII) and do not even take into consideration the possible formation of Mn(IV) (as intermediate phase) during the reduction process as well as the possible formation of non-stoichiometric oxides. In Lines 156-157 the Authors state that an increased volume of DMF “means” a higher pH-value but there is no mention of the pH values. Did the Authors measure it? On the other hand, the data reported in Line 123 are close to the Mn: O = 1: 2 ratio.
Finally in the γ-MnOOH manganite the oxidation state of Mn is 3+ (see as example: ACS Catal. 2016, 6, 3, 2089-2099 (February 16, 2016))
Conclusions
In this section a short summary of the experimental results is usually summarized and commented. This Reviewer has found barely hints of experimental results.
For the mentioned reasons, this manuscript is outright rejected.
Reviewer 2 Report
The manuscript titled "Study on the synthesis of well-shaped Mn3O4 nanooctahedrons and their good performance for lithium ion batteries" is well-organized, and contains all the components.
However, it lacks some originality and depth necessary to justify the publication.
1) In introduction, 2nd paragraph, the authors described the literature review of the relevant field. But it lacks the necessary arguments to build the case for their approach. Providing more contrasting work of the peers to highlight the area where the authors' performed research work could help alleviate that issue. In this case to support their argument "the synthesis of Mn3O4 nanostructures with uniform size to improve their
electrochemical properties is highly desirable."
2) Another limitation was noticed for the originality of the work itself. The hydrothermal process has been utilized by numerous peers in this field to achieve specific morphology of Mn3O4. The manuscript would have benefited from a series of argument to build their case for their route or how their process is different and significant for the readers to learn about.
3) This manuscript would also benefit from a close editing. I found it difficult to follow the author’s argument due to the many stylistic and grammatical errors.
Reviewer 3 Report
The manuscript 712458 entitled “Study on the synthesis of well-shaped Mn3O4 nanooctahedrons and their good performance for lithium ion batteries” had been submitted for publication in the journal Energies. It is devoted to the development of electrode material for Li-ion batteries having advanced characteristics. This class of materials is of great interest and importance nowadays. There are some comments related to the contents of the manuscript.
In the head of the paper the authors should point the type of their paper (Article, Review, Communication, etc) The authors presented in the manuscript the CV data of the studied electrodes at a constant potential scan rate. It is reasonable to present series of CV curves measured at variation of potential scan rate in order to evaluate limiting stage of current determining process. Then it is advisable to analyze peak current values depending on potential scan rate, and try to use for this purpose Randles-Shevchik equation, as it is likely the mass-transport is the limiting stage of the electrochemical process. The authors should plot the dependencies of current density versus square root of potential scan rate of studied system. The linearity of such dependencies will prove the applicability of Randles-Shevchik equation in certain scan rates domain. Also it is reasonable to check the influence of activity coefficients of ions on the diffusion coefficient values. For this purpose, an approach can be applied developed for the solid state ionic transport in the intercalation materials described in the following papers: Russ. J. Electrochem. 55 (2019) 719-737; Ionics 22 (2016) 483-501; Electrochim. Acta 122 (2014) 187-196. It is recommended to the authors to make additional EIS measurements. It is advisable to use for their analysis the equivalent circuit containing elements with clear physical meaning, which fit to this class of materials. The authors can find appropriate circuit in the following papers: J. Electroanal. Chem. 821 (2018) 140-151; Russ. J. Electrochem. 53 (2017) 706-712. This circuit is versatile enough to analyze impedance spectra of various electrochemical systems. It is obligatory to present calculated spectra at the same figure with experimental ones. All impedances should be presented as specific values /cm2, calculated per total working surface of the electrode. For all fitted spectra calculated circuit parameters should be presented and discussed. It is strongly recommended to indicate frequencies of selected points at the figure with the spectra to make a notion regarding frequency domains of each spectra. It is also advisable to evaluate ionic diffusion coefficient in the material using modified Warburg formula, such an example the authors can find in the following papers: Ionics 22 (2016) 483-501; Electrochim. Acta 230 (2017) 479-491. In the latter one the authors can find several approaches to study electrochemical kinetics of the materials able to maintain rapid charge and discharge processes. It is advisable also to compare results obtained using standard and modified Warburg formula. The high rate charge transport is key factor for modern electrode materials. The class of solid state compounds which is called MAX-phases possess extremely high electroconductive properties. Also it improves cycling performance of Li-ion transport. It is reasonable to provide some short review in the Introduction section concerning examples of application of such compounds for enhancement of electroconductive properties of various electrode materials. The authors can find such an example in the following paper: Monatsh Chem 150 (2019) 489-498.
After mentioned modifications the manuscript can be published in the journal Energies.

Reviewer 4 Report
This manuscript suggested experimental process in which Mn3O4 can be manufactured in a simple way. So, I think this study should be published if the following modifications are made.
1. The use of electrodes using Mn3O4 should be clearly stated as to whether they are cathode or anode. 2. The manuscript is need to provide more detail on the process of Mn3O4 being made. 3. Authors should present detail strong point of Mn3O4 in this manuscript than other Mn3O4-based papers. 4. What benefits does the author suggest in the manufacturing process? 5. It should be more persuasive to explain why performance is superior than other papers using Mn3O4.
Round 2
Reviewer 3 Report
The manuscript 712458 entitled “Study on the synthesis of well-shaped Mn3O4 nanooctahedrons and their good performance for lithium ion batteries” had been resubmitted after revision for publication in the journal Nanomaterials. It is devoted to the development of electrode material for Li-ion batteries having advanced characteristics. This class of materials is of great interest and importance nowadays. Evidently, the authors considered all the reviewers’ comments. Now the manuscript can be published in the journal Nanomaterials without further modifications.

Reviewer 4 Report
Through the kind and adequate responses of authors, the doubts are dispelled and the errors of the manuscript are correctly revised. I think the authors provide quite reasonable real time sensor detect experiment with real-time changes.
Thus, I gladly recommend that this manuscript might be suitable to publish to this journal.